Dynamics of a methanol-fed marine denitrifying biofilm: 2—impact of environmental changes on the microbial community

Villemur Richard richard.villemur@iaf.inrs.ca 1
Payette Geneviève 1
Geoffroy Valérie 2
Mauffrey Florian 3
Martineau Christine 4
1 INRS-Centre Armand-Frappier Santé et Biotechnologie , Laval , Québec , Canada
2 Lallemand , Montréal , Québec , Canada
3 Université de Genève , Geneva , Switzerland
4 Laurentian Forestry Centre , Québec , Canada
Gillespie Joseph
Electronic publication date: 2019 Aug 13
Publication date: 2019
Volume: 7
Electronic Location ID: e7467
Received 2019 Apr 17; Accepted 2019 Jul 12
Copyright: ©2019 Villemur et al.
Copyright year: 2019
Copyright holder: Villemur et al.
License: This is an open access article distributed under the terms of the Creative Commons Attribution License, which permits unrestricted use, distribution, reproduction and adaptation in any medium and for any purpose provided that it is properly attributed. For attribution, the original author(s), title, publication source (PeerJ) and either DOI or URL of the article must be cited.
License URL: https://creativecommons.org/licenses/by/4.0/

Keywords: Denitrification, Biofilm, Hyphomicrobium, Methylophaga, Marine environment, Metatranscriptome, Microbial diversity

Funding: Natural Sciences and Engineering Research Council of Canada # RGPIN-2016-06061 This research was supported by a grant to Richard Villemur from the Natural Sciences and Engineering Research Council of Canada # RGPIN-2016-06061. The funders had no role in study design, data collection and analysis, decision to publish, or preparation of the manuscript.

==============================
Background

The biofilm of a methanol-fed, marine denitrification system is composed of a multi-species microbial community, among which Hyphomicrobium nitrativorans and Methylophaga nitratireducenticrescens are the principal bacteria involved in the denitrifying activities. To assess its resilience to environmental changes, the biofilm was cultivated in artificial seawater (ASW) under anoxic conditions and exposed to a range of specific environmental conditions. We previously reported the impact of these changes on the denitrifying activities and the co-occurrence of H. nitrativorans strain NL23 and M. nitratireducenticrescens in the biofilm cultures. Here, we report the impact of these changes on the dynamics of the overall microbial community of the denitrifying biofilm.

Methods

The original biofilm (OB) taken from the denitrification system was cultivated in ASW under anoxic conditions with a range of NaCl concentrations, and with four combinations of nitrate/methanol concentrations and temperatures. The OB was also cultivated in the commercial Instant Ocean seawater (IO). The bacterial diversity of the biofilm cultures and the OB was determined by 16S ribosomal RNA gene sequences. Culture approach was used to isolate other denitrifying bacteria from the biofilm cultures. The metatranscriptomes of selected biofilm cultures were derived, along with the transcriptomes of planktonic pure cultures of H. nitrativorans strain NL23 and M. nitratireducenticrescens strain GP59.

Results

High proportions of M. nitratireducenticrescens occurred in the biofilm cultures. H. nitrativorans strain NL23 was found in high proportion in the OB, but was absent in the biofilm cultures cultivated in the ASW medium at 2.75% NaCl. It was found however in low proportions in the biofilm cultures cultivated in the ASW medium at 0–1% NaCl and in the IO biofilm cultures. Denitrifying bacterial isolates affiliated to Marinobacter spp. and Paracoccus spp. were isolated. Up regulation of the denitrification genes of strains GP59 and NL23 occurred in the biofilm cultures compared to the planktonic pure cultures. Denitrifying bacteria affiliated to the Stappia spp. were metabolically active in the biofilm cultures.

Conclusions

These results illustrate the dynamics of the microbial community in the denitrifying biofilm cultures in adapting to different environmental conditions. The NaCl concentration is an important factor affecting the microbial community in the biofilm cultures. Up regulation of the denitrification genes of M. nitratireducenticrescens strain GP59 and H. nitrativorans strain NL23 in the biofilm cultures suggests different mechanisms of regulation of the denitrification pathway in the biofilm. Other denitrifying heterotrophic bacteria are present in low proportions, suggesting that the biofilm has the potential to adapt to heterotrophic, non-methylotrophic environments.

Introduction

Most naturally-occurring microbial biofilms, such as those encountered in bioremediation processes, are composed of multiple microbial species. Studying such complex biofilms is a challenge, as each species can influence the biofilm development. The biofilm microbial community inside a bioremediation process adapts to the prescribed operating conditions and shapes the efficiency of the bioprocess to degrade the pollutant(s). Usually, the microbial community in such bioprocesses is complex and composed of main degraders but also of secondary microorganisms that could provide benefits to the degraders or could simply contribute to the degradation intermediates or waste. It is recognized that a complex microbial community is more resilient to “unexpected” changes in the operation of the bioprocesses than a single species biofilm, as some of the minor degraders take over the main degraders affected by the changes (Cabrol & Malhautier, 2011; Roder, Sorensen & Burmolle, 2016; Salta et al., 2013; Tan et al., 2017). The mechanisms of how a microbial population in a biofilm adapts to changes, however, are poorly understood.

The Montreal Biodome, a natural science museum, operated a continuous fluidized-bed methanol-fed denitrification reactor to remove nitrate (NO3−) that accumulated in the 3 million-L seawater aquarium. The fluidized carriers in the denitrification reactor were colonized by naturally-occurring multispecies microorganisms to generate a marine methylotrophic denitrifying biofilm estimated to be composed of 15–20 bacterial species (Labbé et al., 2003). The main bacteria responsible of the denitrifying activities belong to the alphaproteobacteria Hyphomicrobium nitrativorans (strain representative NL23) and to the gammaproteobacteria Methylophaga nitratireducenticrescens (strain representative JAM1), both methylotrophs, that accounted for 60–80% of the biofilm (Labbé et al., 2003; Labbé et al., 2007; Martineau et al., 2013b; Villeneuve et al., 2013).

Denitrification takes place in bacterial cells where N oxides serve as terminal electron acceptors instead of oxygen (O2) for energy production when oxygen depletion occurs, leading to the production of gaseous nitrogen (N2). Four sequential reactions are strictly required for the reduction of NO3− to gaseous nitrogen, via nitrite (NO2−), nitric oxide (NO) and nitrous oxide (N2O), and each of these reactions is catalyzed by different enzymes, namely NO3− reductases (Nar and Nap), NO2− reductases (NirS and NirK), NO reductases (Nor) and N2O reductases (Nos) (Kraft, Strous & Tegetmeyer, 2011; Philippot & Hojberg, 1999; Richardson et al., 2001). Whereas H. nitrativorans strain NL23 possesses the four reductases for the complete denitrification pathway, M. nitratireducenticrescens strain JAM1 performs incomplete denitrifying activities, as it lacks a dissimilatory NO-forming nitrite reductase (Auclair et al., 2010; Martineau et al., 2013a; Mauffrey et al., 2017; Mauffrey, Martineau & Villemur, 2015; Villeneuve et al., 2012). Using degenerated PCR primers for the detection of denitrification genes, we showed that there are probably other denitrifying bacteria in the biofilm, one to four orders of magnitude lower in proportions than M. nitratireducenticrescens strain JAM1 and H. nitrativorans strain NL23 (Auclair, Parent & Villemur, 2012). These other bacteria may play a role if the bioprocess undergoes stress conditions or changes in the operation mode.

We have initiated a study with the aim of assessing the performance of the Biodome denitrifying biofilm subjected to environmental changes. The original biofilm (OB) taken from the Biodome denitrification system was cultivated in an artificial seawater (ASW) under batch mode and anoxic conditions at laboratory scale and exposed to a range of specific physico-chemical parameters. Such parameters included a range of NaCl, NO3−, NO2− and methanol concentrations, and varying pH and temperature. These parameters were chosen as possible factors that could affect a denitrification reactor. Thus, the objectives of this study were to determine the impact of these changes: (1) on the denitrification performance of the biofilm; (2) on the dynamics of the co-occurrence of H. nitrativorans and M. nitratireducenticrescens in the biofilm; and (3) on the overall microbial community. The fourth objective of the study was to determine whether denitrifying bacteria other than H. nitrativorans strain NL23 and M. nitratireducenticrescens strain JAM1 are present in the biofilm.

Results for the first two objectives and partially the fourth objective were reported by Geoffroy et al. (2018) and Payette et al. (2019). We showed that the denitrifying biofilm can sustain denitrifying activities in most of the tested conditions. Inhibition occurred when these biofilm cultures were exposed to high pH (10) or to high methanol concentrations (1.5%). The highest specific denitrification rates occurred when the biofilm cultures were cultivated at 64.3 mM NO3− and 0.45% methanol (C/N = 1.5), and at 30 ° C. Poor biofilm development occurred in biofilm cultures cultivated at 5% and 8% NaCl. We also showed that the NaCl concentrations in the ASW medium have significant impacts on the population of H. nitrativorans strain NL23, with its displacement by a subpopulation of the species M. nitratireducenticrescens (strain GP59 as representative), which can perform the complete denitrification pathway.

Results for the third and fourth objectives are presented here. The composition of the bacterial community of the different biofilm cultures was determined by sequencing the 16S ribosomal RNA (rRNA) genes. A culture dependent approach was used to recover new denitrifying bacterial isolates from the biofilm cultures. To complement these two objectives, we derived the metatranscriptome from selected biofilm cultures. These metatranscriptomes were analyzed to determine the composition of the active microbial community in the biofilm cultures but also to assess their metabolic contributions, such as those involved in denitrification. Furthermore, metatranscriptomic analyses provided further indications on the dynamics of H. nitrativorans and M. nitratireducenticrescens in these cultures (second objective) by assessing changes in metabolic pathways of H. nitrativorans strain NL23 and M. nitratireducenticrescens strain GP59 between the planktonic pure cultures and the biofilm cultures. Our study is the first that give a comprehensive picture of the microbial community of a methylotrophic denitrifying biofilm and its adaptation to specific changes.

Material and Methods

Cultivation of the original biofilm to different culture conditions

The formulations of the artificial seawater (ASW) medium and the commercial Instant Ocean (IO) medium (Table S1), and the different conditions of the biofilm cultures were described by Payette et al. (2019). Briefly, the biomass of several carriers taken from the denitrification reactor of the Montreal Biodome was scraped, dispersed, then distributed to several vials containing twenty free carriers and 60 mL prescribed medium (Table 1; Fig. S1). The vials were incubated under anoxic conditions at 23 °C or 30 °C (Table 1) and shaken at 100 rpm (orbital shaker). In average once a week, the twenty carriers were taken, gently washed to remove the excess medium and the planktonic bacteria, then transferred into fresh anoxic medium and incubated under the same conditions (Fig. S1). The Ref300N-23C biofilm cultures (for 300 mg NO3−-N/L, 23 °C) were defined as the reference biofilm cultures. These cultures were used by Payette et al. (2019) as a reference to compare results between the different culture conditions. The protocols to measure NO3− and NO2− concentrations, and to extract DNA from the biofilm cultures or the planktonic pure cultures were described in Payette et al. (2019) and Geoffroy et al. (2018).

Table 1 Biofilm culture conditions.

Name	Medium	NO3−	Methanol	NaCl	Temp	Specifica denitrification rates	
		mM (mg-NO3-N/L)c	% (v/v)c	% (w/v)	°C	mM-NOx h−1 mg-protein−1	
Ref300N-23Cb	ASW	21.4 (300)	0.15	2.75	23	0.0530	
300N-30C	ASW	21.4 (300)	0.15	2.75	30	0.0946	
900N-23C	ASW	64.3 (900)	0.45	2.75	23	0.0637	
900N-30C	ASW	64.3 (900)	0.45	2.75	30	0.0979	
0%NaCl	ASW	21.4 (300)	0.15	0	23	0.0911	
0.5%NaCl	ASW	21.4 (300)	0.15	0.5	23	0.0712	
1%NaCl	ASW	21.4 (300)	0.15	1.0	23	0.0357	
IO	IO	21.4 (300)	0.15	3.0d	23	0.0611	
Notes.

The original biofilm was cultured in triplicates in these conditions at pH 8.0. The carriers were transferred 5 times in fresh medium around each week before measuring the denitrifying activities.

a From Payette et al. (2019).

b Reference biofilm cultures.

c The C/N ratio was 1.5 in all biofilm cultures.

d The exact amount of NaCl added in the IO medium is not known. See Payette et al. (2019) for the IO composition. For comparison, the amount of Na+ and Cl− in the ASW medium is 3.2%.

In gray are changed parameters from the reference biofilm cultures.

IO Instant Ocean medium

16S rRNA gene analysis

DNA extracted from triplicate biofilm cultures was pooled before sequencing. Total DNA samples from seven biofilm cultures (Table 1; Ref300N-23C, 300N-30C, 900N-23C, 900N-30C, 0%NaCl, 0.5%NaCl and 1.0%NaCl) were sent to the sequencing service at the Research and Testing Laboratory (RTL, Lubbock, Texas, USA). A region of the 16S rRNA gene was PCR amplified using the 28F-519R primers (5′ GAGTTTGATCNTGGCTCAG 3′ and 5′ GTNTTACNGCGGCKGCTG 3′, covering the V1–V2–V3 variable regions) and subjected to pyrosequencing using a Roche 454 FLX genome sequencer system. The sequencing service (RTL) performed denoising and chimera analyses (details provided in Document S1). The high-quality reads were then processed in the RDP pipeline at the Ribosomal data project (RDP) web site (Cole et al., 2014). Reads were clustered into operational taxonomic units (OTU) using a 97% identity threshold. DNA extracted from the OB (from frozen stock) and from fresh IO biofilm cultures (Table 1) were sent to the sequencing service of Genome Quebec Innovation Center (Montreal, QC, Canada). In these cases, the 16S rRNA sequences covering the V6–V7–V8 variable regions (5′ ACACTGACGACATGGTTCTACA 3′ and 5′ TACGGTAGCAGAGACTTGGTCT 3′) were PCR amplified and sequenced by Illumina MiSeq PE250 (250 bp paired-end sequencing reactions). The reads were processed based on Peck et al. (2016). Briefly, paired-end reads were merged with minimum and maximum overlap length between the two reads of 20 and 250 bases, respectively, with 30% mismatched bp tolerance in the overlap region. The merged reads were processed using the software UPARSE (Edgar, 2013). Sequences were truncated to a uniformized length to 420 bp. Reads with a low-quality score were removed using 2.0 as the maximum expected error value. The high-quality reads were de-replicated, sorted by size and singletons were removed. The resulting reads were clustered into operational taxonomic units (OTU) with the UPARSE-OTU clustering method using a 97% identity threshold. Chimeric OTU were removed by UPARSE-REF algorithm and with the software UCHIME ran against ChimeraSlayer ‘gold’ reference database (Edgar et al., 2011). All representative sequences of the OTUs (from pyrosequencing and Illumina) were checked again for chimeras with the DECIPHER v 2.0 program (http://www2.decipher.codes/FindChimeras.html) (Wright, Yilmaz & Noguera, 2012). The affiliation of the OTUs to the most probable genus was determined by the CLASSIFIER program at the RDP web site (Document S2). 16S rRNA sequence reads were deposited in the GenBank Sequence Read Archive (SRA) under the accession number PRJNA524642. Principal component analysis of the proportion of reads associated to the bacterial profiles was performed at ClustVis web site (https://biit.cs.ut.ee/clustvis/) (Metsalu & Vilo, 2015).

Isolation of bacterial isolates

Biofilm of the Ref300C-23C biofilm cultures was scraped from the carriers and dispersed in saline solution (3% NaCl, 34.2 mM phosphate buffer pH 7.4), and serial dilutions were made and inoculated onto these agar plate media: (1) R2A medium (complex organic carbons; EMD Chemicals Inc., Gibbstown, NJ, USA), (2) Marine Agar 2216 (marine medium with yeast extract and peptone as carbon source; Becton, Dickinson and Co., Sparks, MD, USA), (3) Methylophaga medium 1403 (American Type Culture Collection [ATCC], Manassas, VA, USA) and (4) the ASW medium; these two latter media were supplemented with 1.5% agar and 0.3% v/v methanol. The isolation procedure, the taxonomic affiliation of the isolates and the measurement of their denitrifying activities were carried out as described by Geoffroy et al. (2018). The 16S rRNA gene sequences were deposited in GenBank under the accession numbers MK571459–MK571476.

Transcriptomes

Planktonic pure cultures of M. nitratireducenticrescens strains JAM1 and GP59 were performed in the Methylophaga 1403 medium and of H. nitrativorans strain NL23 in the 337a medium as described by Martineau, Mauffrey & Villemur (2015) and Mauffrey, Martineau & Villemur (2015). These cultures were carried out in triplicate with methanol (0.3%) and NO3− (21.4 mM [300 mg-N/L]) under anoxic conditions at 30 °C. The biomass was collected by centrifugation when the NO3− reduction was near completion, and total RNA was extracted as described by Mauffrey, Martineau & Villemur (2015). For the biofilm cultures, at the end of the the fifth transfer cultures, the biomass of each replicate was scraped from carriers and used to extract total RNA. The RNA samples were sent to the sequencing service for RNA sequencing (RNAseq) by Illumina (Genome Quebec Innovation Center, Montreal QC, Canada). Because of limited amount of biofilm available, total RNA from the triplicate biofilm samples were pooled before sending to the sequencing service. For the planktonic pure cultures, RNAseq was performed on each replicate. The Ribo-Zero™ rRNA Removal Kit (Meta-Bacteria; Epicentre, Madison, WI, USA) was used to deplete total RNA of the ribosomal RNA. The RNA was then treated with the TruSeq Stranded mRNA Sample Prep Kit (Illumina Inc, San Diego, CA, USA).

All computations were made on the supercomputer Briarée from the Université de Montréal, managed by Calcul Québec and Compute Canada. Raw reads were filtered to remove low quality reads using FASTX toolkit (http://hannonlab.cshl.edu/fastx_toolkit/) by discarding any reads with more than 10% nucleotides with a PHRED score <20. The resulting reads from each sample/replicate were aligned respectively to the genome of M. nitratireducenticrescens strain JAM1 (GenBank accession number CP003390.3), to the genome and plasmids of M. nitratireducenticrescens strain GP59 (CP021973.1, CP021974.1, CP021975.1) and to the genome of H. nitrativorans strain NL23 (CP006912.1) using Bowtie (v 2.2.3) with default parameters. SAMtools (v 0.1.18) and BEDtools (v 2.20.1) were used for the generation of sam and bam files, respectively. Significance for difference in the relative transcript levels of a gene (defined as transcript per million: TPM) between planktonic pure cultures and biofilm cultures was performed with the R Bioconductor NOIseq package v2.14.0 (NOIseqBio) (Tarazona et al., 2011) and run with the R software v3.2.3 (Team, 2015). Because the RNAseq from the biofilm samples were derived from one pooled RNA preparation, the “no replicate parameter” was set (pnr = 0.2, nss = 5 and v = 0.02; pseudoreplicate generated) in NOIseq as described by Tarazona et al. (2011) under the NOISeq-sim section. Briefly, NOISeq-sim assumes (quoting) ”that read counts follow a multinomial distribution, where probabilities for each gene in the multinomial distribution are the probability of a read to map to that gene”. Results from this statistical analysis showed that genes that had at least >2-fold higher transcript levels from one type of cultures to the other showed significant differences. RNAseq reads from the planktonic pure cultures and the biofilm cultures were deposited in the SRA under the accession number PRJNA525230. Annotations were based on services provided by GenBank (https://www.ncbi.nlm.nih.gov/genbank), RAST (Rapid Annotation using Subsystem Technology; http://rast.nmpdr.org) and KEGG (Kyoto Encyclopedia of Genes and Genomes; https://www.genome.jp/kegg) (Document S3).

To derive transcript reads not associated to M. nitratireducenticrescens and H. nitrativorans, reads were aligned to a concatenated sequence consisting of the three reference genomes (JAM1 + GP59 + NL23) and the two plasmids (from strain GP59). The reads that did not align were kept. Unaligned reads were de novo assembled at the National Center for Genome Analysis web site (https://galaxy.ncgas-trinity.indiana.edu) by Trinity v. 2.4.0 (Grabherr et al., 2011). These transcripts were deposited in SRA under the accession number PRJNA525230. Estimation of the transcript abundance of the de novo assembled sequences was performed by RSEM (Li & Dewey, 2011). The resulting assembled sequences were annotated at the Joint Genomic Institute (https://img.jgi.doe.gov/cgi-bin/m/main.cgi) to find open reading frames with their putative function and affiliation (GOLD Analysis Project Id: Ga0307915, Ga0307877, Ga0307760). The annotations were then verified manually for discrepancies within the assembled sequences (Documents S4–S7).

Results

Bacterial composition of the biofilm cultures by 16S rRNA gene sequencing

As reported by Payette et al. (2019), the original biofilm (OB) collected from the Biodome denitrification reactor was used as inoculum to colonize new carriers in a series of anoxic biofilm cultures cultivated under different conditions (Table 1; Fig. S1). We selected eight of these biofilm cultures for our present analysis for the following reasons. The Ref300N-23C biofilm cultures were used as reference for comparison analysis. The 300N-30C, 900N-23C and 900N-30C biofilm cultures had higher specific denitrification rates compared to the Ref300N-23C biofilm cultures. The 0%, 0.5% and 1.0% NaCl ASW biofilm cultures and the IO biofilm cultures were chosen because of the persistence of H. nitrativorans NL23 in these cultures as opposed to the other cultures. The composition of the bacterial community of these biofilm cultures and the OB was determined by sequencing the 16S rRNA genes to assess the impact of these specific conditions on the bacterial community (Fig. 1A; Table 2). The bacterial profiles of the OB and the IO biofilm cultures were distinct to each other, and to the seven biofilm cultures cultivated with different formulations in the ASW medium (Fig. 1B). The bacterial profiles of these latter cultures were however not very different because of the high proportions of Methylophaga spp. (>85%).

Figure 1 Proportion of affiliated OTUs in the biofilm cultures.

(A) Bacterial composition of OB and the IO biofilm cultures was determined by sequencing the V6–V7–V8 variable regions of the 16S rRNA gene by Illumina, whereas the other samples were determined by sequencing the V1–V2–V3 variable regions by pyrosequencing. (B) Principal component analysis of the bacterial profiles of the biofilm cultures and the OB listed in Table 2.

In the OB, high proportions of the 16S rRNA gene sequences were related to Hyphomicrobium spp. (45.8%) followed by Oceanibaculum spp. (12.3%), Aquamicrobium spp. (11.6%); Methylophaga spp. accounted for 3.5% (Table 2). In the IO biofilm cultures, the proportion of Methylophaga spp. was 12 times higher (42.8%) than in the OB, whereas it was 5.5 times lower for Hyphomicrobium spp. (8.4%). Higher proportions of Marinicella spp. (7.2%) and Winogradskyella spp. (4.3%) along with a much lower proportion of Aquamicrobium spp. (0.44%) were observed in the IO biofilm cultures compared to the OB (Table 2).

In the biofilm cultures cultivated with the four combinations of NO3−/methanol concentrations and temperatures in ASW medium containing 2.75% NaCl (Ref300N-23C, 300N-30C, 900N-23C, 900N-30C), Methylophaga spp. accounted for >90% of the 16S rRNA gene sequences followed by Marinicella spp. with proportions ranging from 1.5% to 5.0% (Fig. 1; Table 2). No sequences were found affiliated to Hyphomicrobium spp. under these conditions. In the biofilm cultures cultivated at low NaCl concentrations (0% NaCl, 0.5% NaCl, 1% NaCl), Hyphomicrobium spp. accounted for 11.8%, 6.8% and 0.25%, respectively of the 16S rRNA gene sequences (Fig. 1; Table 2). Methylophaga spp. was still the dominant genus with more than 85% of the 16S rRNA gene sequences; 16S rRNA gene sequences affiliated to Marinicella spp. were also found in significant proportions (Fig. 1; Table 2). Finally, 16S rRNA gene sequences affiliated to Stappia spp. were found in all biofilm cultures and in the OB.

The 16S rRNA gene sequences from the OB and the IO biofilm cultures that were derived by Illumina sequencing generated several thousands of reads affiliated to Hyphomicrobium spp. and Methylophaga spp. (Table 3). This tremendous amounts of sequences allowed assessing the presence of species other than H. nitrativorans and M. nitratireducenticrescens in these two biofilms. The phylogenic analyses performed on these sequences allowed regrouping the OTUs in three clusters for Hyphomicrobium spp., and also three clusters for Methylophaga spp. (Figs. S2A and S2B). The vast majority (>90%) of the 16S rRNA gene sequences associated to these OTUs were affiliated to H. nitrativorans or M. nitratireducenticrescens, respectively, in the OB and the IO biofilm cultures (Clusters 1, Table 3). A small proportion of the OTUs (clusters 2 and 3) were affiliated to other Hyphomicrobium or Methylophaga, which suggests that other members of these genera were present in these biomasses in very low proportions.

Isolation of denitrifying bacterial isolates from the biofilm cultures

The biomass of the Ref300N-23C biofilm cultures was dispersed on different nutrient agar plates to isolate denitrifying bacteria other than H. nitrativorans strain NL23 and M. nitratireducenticrescens strain GP59. Isolates affiliated to the genera Marinobacter, Pseudomonas, Paracoccus, Roseovarius, Thalassobius, Winogradskyella, Aequorivita and Exiguobacterium (Table 4) were recovered from the Marine medium 2216 plates, which contains yeast extract and peptone (Atlas, 1993). Only isolates affiliated to the genera Marinobacter and Paracoccus showed consumption of NO3− and NO2− and production of gas, suggesting that they possess the complete denitrification pathway. The three isolates affiliated to the Paracoccus spp. have identical 16S rRNA sequences with the one of Paracoccus sp. strain NL8 that was isolated from the Biodome denitrification reactor (Labbé et al., 2003). This result suggests that strain NL8 persisted in the biofilm cultures. One representative of Paracoccus isolates (GP3) could grow with methanol as sole source of carbon; Marinobacter sp. GP2 could not.

Table 2 Most probable affiliation of 16S rRNA gene sequences in the biofilm cultures and the original biofilm.

Affiliation	Biofilm cultures (proportion %)	
			OB	IO	Ref300N-23C	900N-23C	300N-30C	900N-30C	0%NaCl	0.5%NaCl	1.0%NaCl	
Ignavibacteriae; Igavibacteriales;	Ignavibacterium	0.37	0.13		0.01	0.19	0.06				
Tenericutes; Acholeplasmatales;	Acholeplasma			0.05	0.04	0.05		0.04	0.52	0.02	
Bacteroidetes; Flavobacteriales;	Aequorivita	0.01	0.18		0.10	0.14	0.04	0.01	0.11	0.22	
”	”	Muricauda	0.06	0.08			0.01	0.01		0.10		
”	”	Winogradskyella	0.28	4.28	0.10	0.11	0.15	0.11		0.20	0.11	
Proteobacteria;												
Alphaproteobacteria; Rhizobiales;	Aminobacter	0.33	0.01		0.05				0.22		
”	”	Aquamicrobiuma	11.6	0.46					0.02			
”	”	Hoeflea	0.30	0.76	0.03	0.01			0.06	0.12	0.06	
”	”	Nitratireductor	1.40	0.03								
”	”	Hyphomicrobiuma	45.8	8.40					10.8	6.78	0.25	
	H. nitrativorans NL23, qPCR (cp napA/ng)	8.7(7.2)a104	7.0(4.3)a104	1.8(1.8)a102	6.0(3.2)a101	1.3(0.5)a102	1.3(0.9)a102	2.8(1.0)a104	5.3(6.5)a104	1.1(0.9)a104	
Alphaproteobacteria; Rhodobacterales;	Litorisediminicola	0.38	1.43								
”	”	Lutimaribacter	1.71	8.93								
”	”	Marinovum	1.28	1.62								
”	”	Maritimibacter	2.50	2.44		0.01				0.01		
”	”	Oceanicola	0.06	0.01			0.08	0.03		0.01		
”	”	Paracoccusa			0.01	0.07	0.01			0.02		
”	”	Roseovarius	7.32	8.18					0.01			
”	”	Stappiaa	0.02	1.26	0.09	0.43	0.27	0.42	0.27	0.35	0.17	
Alphaproteobacteria; Rhodospirillales	Oceanibaculum	12.3	7.36		0.03	0.01	0.01		0.20	0.04	
Proteobacteria;												
Gammaproteobacteria; Alteromonadales	Marinobactera	0.15	0.05	0.06	0.05	0.02			0.01	0.01	
”	Oceanospirillales	Marinicella	0.78	7.24	1.46	1.50	3.40	5.02	0.23	1.55	0.94	
”	Pseudomonadales	Pseudomonasa		0.09	0.11	0.04	0.13	0.52	0.64	0.80	0.05	
”	Thiotrichales	Methylophaga	3.50	42.8	97.7	97.1	95.1	91.8	85.7	88.3	97.9	
	M. nitratireducenticrescens, qPCR (cp narG1/ng)	5.6(3.2)a103	5.6(4.6)a104	2.3(0.6)a105	1.9(0.5)a105	2.1(0.4)a105	2.1(1.4)a105	4.7(2.0)a104	3.6(2.7)a104	1.5(0.2)a105	
Others			9.86	4.27	0.25	0.37	0.53	2.00	2.28	0.69	0.25	
Total of reads			348,358	319,265	9,610	14,048	8,532	13,915	13,800	9,375	12,560	
Notes.

The V1–V3 regions of the 16S rRNA gene sequences were amplified and sequenced by pyrosequencing except for the OB and IO samples from which the V6–V8 regions were sequenced by Illumina (see Document S2 for complete analysis and sequences).

OB Original biofilm

In grey: qPCR results are from Payette et al. (2019), with standard deviation values under parentheses of triplicate cultures. narG1 used in qPCR for M. nitratireducenticrescens cannot discriminate strain JAM1 and strain GP59 (identical sequences between the two).

a Identified genus with species that were reported involved in denitrification.

Table 3 16S Operational Taxonomic Units (OTUs) affiliated to Hyphomicrobium spp. and Methylophaga spp. in the OB and the IO biofilm cultures.

		Number of OTUs	Number reads	Proportion of reads %	
			OB	IO	OB	IO	
Sequence identity with H. nitrativorans							
Cluster 1	95–100%	12	147,209	26,239	92.3	97.8	
Cluster 2	91–97%	5	10,779	530	6.8	2.0	
Cluster 3	89–92%	3	1,537	51	0.96	0.19	
Sequence identity with M. nitratireducenticrescens							
Cluster 1	94–100%	18	11,809	136,197	96.4	99.7	
Cluster 2	92.9–93.6%	3	321	284	2.62	0.21	
Cluster 3	93.4–94.1%	3	115	65	0.94	0.05	
Notes.

Cluster classification is based on phylogenic analyses of the 16S rRNA gene sequences affiliated to H. nitrativorans and M. nitratireducenticrescens (see Tables S1 and S2, and Figs. S2A and S2B).

Table 4 Affiliation of the isolates isolated from the reference biofilm cultures (Ref300N-23C).

Isolates	Denitrifying activitiesa	
Alphaproteobacteria, Rhodobacterales		
Rhodobacteraceae GP11	No	
Paracoccus sp. GP3, GP8, GP20	Full	
Roseovarius sp. GP9, GP10, GP13, GP14	No	
Roseovarius sp. GP12	No	
Thalassobius sp. GP19	No	
Gammaproteobacteria		
Marinobacter sp. GP1, GP2, GP24	Full	
Pseudomonas GP41	Nitrate	
Bacteroidetes, Flavobacteriales		
Aequorivita sp. GP15	No	
Winogradskyella sp. GP16, GP18	No	
Bacillales		
Exiguobacterium sp. GP46	No	
Notes.

a Full: nitrate and nitrite are consumed. Nitrate: only nitrate was consumed.

Metatranscriptomic analysis of the biofilm cultures

The metatranscriptomic approach has allowed assessing the contributions of the microbial community to the metabolic processes in the biofilm cultures. We have chosen to focus on three biofilm cultures, which were the Ref300N-23C (the reference biofilm cultures), 900N-30C (highest denitrification rates; Table 1) and 0% NaCl (persistence of H. nitrativorans strain NL23) biofilm cultures. Because the genomes of H. nitrativorans strain NL23 and M. nitratireducenticrescens strain JAM1 and strain GP59 were available, we first determined changes in the transcript levels of genes associated to these genomes between the biofilm cultures and the planktonic pure cultures of the respective strains. To assess the contribution of other microorganisms in biofilm cultures, the reads from the metatranscriptomes that did not align with the three reference genomes were used to derive de novo assembled transcripts. These transcripts were annotated for function and bacterial affiliation.

Gene expression profiles of M. nitratireducenticrescens strain GP59 in the biofilm cultures

Because >80% of the genomes of strains JAM1 and GP59 are identical, high proportions of reads from the biofilm metatranscriptomes can align to both genomes. Geoffroy et al. (2018) showed that the gene expression profiles of the common genes between both strains in planktonic pure cultures were similar. In Payette et al. (2019), the concentrations of strain GP59 and strain JAM1 in the biofilm cultures were determined by qPCR. In the three selected biofilm cultures for the metatranscriptomic analysis, the concentrations of strain GP59 (copies of nirK by ng biofilm DNA) were one to three orders of magnitude higher than those of strain JAM1 (copies of tagH by ng biofilm DNA). Because of these differences, it was assumed that most of the transcript reads associated to M. nitratireducenticrescens in the biofilm cultures were from strain GP59. Metatranscriptomic analysis in relation with strain JAM1 is described in Documents S8 and S11. The transcriptome of strain GP59 was also derived from planktonic pure cultures cultivated under anoxic conditions in the Methylophaga 1403 medium (Fig. S1). The choice of this medium was because suboptimal growth occurred with strain GP59 in ASW medium. The relative transcript levels of the corresponding genes in the biofilm cultures and the planktonic pure cultures were compared to assess changes in the metabolisms of the strain that occurred between the two environments. All quantitative changes described below of the transcript levels in the biofilm cultures are expressed relative to the transcript levels in the planktonic pure cultures.

Among all genes of strain GP59, between 11% and 21% of them had higher relative transcript levels in the biofilm cultures. At the opposite, 6 to 17% of all genes of strain GP59 were expressed at higher relative transcript levels in planktonic pure cultures (Fig. 2). Strain GP59 contains two plasmids, and most of the genes encoded by these plasmids had much lower relative transcript levels in the biofilm cultures (Fig. 2). Genes involved in the nitrogen metabolism and iron transport were globally at higher relative transcript levels in the biofilm cultures (Table 5; Figs. 2 and 3).

Figure 2 Relative expression profiles of M. nitratireducenticrescens GP59 and H. nitrativorans NL23 in biofilm cultures.

All the deduced amino acid sequences associated to the genome and plasmids of strain GP59 and the genome of strain NL23 were submitted to the BlastKOALA (genome annotation and KEGG mapping) at the Kyoto encyclopedia of genes and genomes (KEGG). Genes associated to specific metabolisms were sorted out and the corresponding ratios of the Biofilm Transcripts Per Million (TPM) versus the pure culture TPM were derived. When the ratios were <1, the negative inverse values (−1/ratio) were calculated. Data are expressed as the percentage of genes in each category that are more expressed in the biofilm cultures (right, 2–5 times, and > 5 times) or in pure cultures (left, −2 to −5 times and > −5 times). Number within parentheses is the number of genes involved in the selected pathways. Other metabolic profiles are detailed in Figs. S3 and S4.

Table 5 Changes in the relative transcript levels of genes involved in the nitrogen metabolism in strain GP59 and strain NL23.

Genes	Ratio TPM Biofilm cultures/TPM pure cultures	
	GP59	NL23	
	Ref300N-23C	900N-30C	0%NaCl	0%NaCl	Genes	
Denitrification						
narXL	ns	ns	ns	−9.4	napGH	
narK1K2GHJI-1	ns	−7.4	2.1	8.2	napEFDABC	
narK12	ns	ns	2.0			
narGHJI-2	2.7	2.8	3.0			
nirK	11	4.7	ns	49	nirKV	
norCBDQ-1	ns	ns	−2.7	5.9	norCBQDE	
norRE	ns	ns	−3.1			
norCBQD-2	2.4	3.8	2.8			
nosRZDFYL	ns	ns	ns	5.2	nosRZDFYLX	
nnrS (3)	ns	ns	ns	ns	nnrS (2)	
nsrR (2)	2.1	2.5	ns	ns	nsrR	
DnrN	2.3	2.8	−3.0	ns	nnrU	
NOdiox (2)	ns	ns	ns	4.0	nnrR	
						
Nitrogen assimilatory pathway						
				ns	nasTS	
NO3/NO2 trp	6.0	7.2	17.9	7.8	ntrABC	
nasAnirBD	12	24	42	3.3	nirBAnasA	
ntrYX	ns	ns	ns	ns	ntrYX	
glnA	2.0	3.1	5.1	19	glnA	
				−2.1	glnA (3)	
gltBD: GOGAT	ns	ns	ns	ns	gltBD	
glnB	ns	ns	ns	14	glnB	
GDH	ns	ns	ns	ns	GDH	
glnD	ns	ns	ns	−2.9	glnD	
glnE	ns	ns	ns	−3.2	glnE	
glnK	18	24	25	22	glnK	
NH4 trp	11	11	11	20	NH4-trp	
glnLG	ns	2.1	5.9	ns	glnLG	
Notes.

Data values are the ratios of the biofilm-culture TPM divided by the planktonic pure-culture TPM. When the ratios were <1, the negative inverse values (−1/ratio) were calculated. Positive ratios mean higher relative transcript levels in the biofilm cultures, and negative ratios higher relative transcript levels in the planktonic pure cultures.

ns no significant changes

glnA glutamine synthetase

gltBD glutamate synthase

GDH glutamate dehydrogenase

glnB nitrogen regulatory protein P-II 1

glnK nitrogen regulatory protein P-II 2

glnD uridylyltransferase

glnLG nitrogen regulation sensor histidine kinase and response regulator

trp transporter

Diox dioxygenase

nnrS involved in response to NO

nsrR NO-sensitive transcriptional repressor

DnrN NO-dependent regulator

nasAnirBD assmilatory nitrate and nitrite reductase

ntrXY Nitrogen regulation proteins

narXL nitrate/nitrite sensor histidine kinase and response regulator

narK nitrate/nitrite transporter

nosRE NO-reductase transcription regulator and activation protein

Figure 3 Relative gene expression profiles of selected metabolic pathways of M. nitratireducenticrescens strain GP59 in the biofilm cultures.

The pathways are based on functions deduced by the annotations (provided by KEGG BlastKoala, RAST and GenBank). The arrow thickness is proportional to the value of the ratio of the biofilm-culture TPM divided by the planktonic pure-culture TPM. The blue arrows represent genes with at least 2-fold lower relative transcript levels in the biofilm cultures. The red arrows represent genes with at least 2-fold higher relative transcript levels in the biofilm cultures. The black arrows represent no changes between both types of cultures in the relative transcript levels. The two-component systems and the transporters that are illustrated in blue are encoded by strains GP59 and NL23. See Documents S9 and S10 for gene description.

For the denitrification genes, narXL encoding the regulatory factors of the nar systems showed no differences between the biofilm cultures and the planktonic pure cultures in the relative transcript levels (Table 5). Small upregulation of the nar2 operon with about 3-fold increases in relative transcript levels occurred in the biofilm cultures. These levels were lower in the 900N-30C biofilm cultures for the nar1 operon and were about the same levels in the two other biofilm cultures and the planktonic pure cultures. The nor1 operon had the same relative transcript levels in the 300N-23C and 900N-30C biofilm cultures and the planktonic pure cultures, and a 3-fold decrease was noticed in these levels in the 0% NaCl biofilm cultures (Table 5). No significant changes in the expression of the nos operon occurred between both types of cultures. The relative transcript levels of nirK were 5 to 10-times higher in the 300N-23C and 900N-30C biofilm cultures, whereas these levels were similar in the 0% NaCl biofilm cultures and the planktonic pure cultures (Table 5). Higher relative transcript levels of genes involved in the ammonium transport and the assimilatory NO3−/NO2− reduction pathway were observed in the biofilm cultures (Table 5; Fig. 3). Absence of nitrogen source other than NO3− in the ASW medium and presence of 37 mM NH4+ in the medium used for the planktonic pure cultures (Methylophaga 1403) could explain these differences in the assimilatory pathway.

Figure 3 illustrates changes in the relative expression profiles in the biofilm cultures of major pathways in strain GP59. The relative transcript levels of mxaFJGI encoding the small and large subunits of the methanol dehydrogenase (MDH) and the cytochrome c-L increased by 2–6 folds in biofilm cultures. Two out of the four mxaF-related products (xoxF) showed 2- to 9-fold decreases in their relative transcript levels in the biofilm cultures. As observed in Methylorubrum extorquens, the genome of M. nitratireducenticrescens strain GP59 encodes three formate dehydrogenase with the same gene arrangement (Chistoserdova et al., 2004). The fdhCBAD operon that encodes the NAD-linked, Mo-formate dehydrogenase showed ca. 60-fold increases in the relative transcript levels in the biofilm cultures, whereas the two other fdh operons stayed at the same levels of those of the planktonic pure cultures. Genes encoding NAD-dependent formate dehydrogenase were upregulated in biofilms formed by Desulfovibrio vulgaris compared to planktonic cultures (Clark et al., 2012). This was also the case in biofilms formed by Staphylococcus aureus where the NAD-dependent formate dehydrogenase genes were among the highest upregulated genes (Resch et al., 2005). In both studies, this upregulation correlated with increase in formate dehydrogenase activity. Contrary to planktonic cultures, accumulation of formate could occur in cell vicinity in the biofilm that would be toxic for the cells (Resch et al., 2005). Therefore, upregulation of fdhCBAD operon could be related to detoxification. The relative transcript levels of the gene encoding the cytochrome c555 were 40–70 times higher in the biofilm cultures. Genes encoding the other cytochromes, pseudoazurins and azurin were expressed at similar levels in both types of cultures. Genes involved in the formaldehyde metabolism to formate and CO2, glycolysis, the ribulose monophosphate pathway, the Entner Doudorof pathway, the tricarboxylic acid cycle, and the pentose pathway showed their relative transcript levels in general unchanged. Few genes in these pathways had 2- to 11-fold differences between the planktonic pure cultures and the biofilm cultures. The genome of strain GP59 encodes the major enzymes involved in the Calvin-Benson-Bassham cycle: the ribulose-bisphosphate carboxylase (Rubisco) and the phosphoribulokinase (Prk). In the 0% NaCl biofilm cultures, the relative transcript levels of the Rubisco gene operon (rbcSL) jumped by 66 times. This upregulation was less pronounced in the Ref300N-23C biofilm cultures (3-fold increase). For the prk gene, the relative transcript levels were 3 to 4 times higher in the 0% NaCl biofilm cultures. The nature of this upregulation is unknown. Except for the cytochrome c555, the relative transcript levels of genes encoding for the oxidative phosphorylation metabolism were unchanged. Several genes involved in iron transport showed higher relative transcript levels in the biofilm cultures (2- to >50-fold increases). The nature of this upregulation in the biofilm cultures is unknown, as the biofilm and planktonic pure cultures were cultivated with trace elements containing iron (Payette et al., 2019).

Gene expression profiles of H. nitrativorans strain NL23 in the 0% NaCl biofilm cultures

As with M. nitratireducenticrescens strain GP59, the transcript levels of genes associated to H. nitrativorans strain NL23 were compared between the biofilm cultures and the planktonic pure cultures to assess changes in the metabolisms that occurred between the two environments. The planktonic pure cultures of strain NL23 were cultivated under anoxic conditions in the 337a medium (Fig. S1). The choice of this medium was because strain NL23 could not grow in ASW (with 2.75% NaCl).

The overall analysis of the three metatranscriptomes confirmed the results obtained by qPCR assays (Payette et al., 2019) and the 16S rRNA gene analysis (Table 2). High number of reads (40 × 106) derived from the metatranscriptome of the 0% NaCl biofilm cultures aligned with the NL23 genome, but <20,000 reads derived from the Ref300N-23C and 900N-30C metatranscriptomes did. In the 0% NaCl biofilm cultures, <10% of all NL23 genes had a higher relative transcript levels in the 0% NaCl biofilm cultures, whereas this was the case for >40% genes in planktonic pure cultures (Fig. 2). These results suggest important changes had occurred in the regulation of gene expression between the planktonic pure cultures and the biofilm cultures. Genes involved in the energy (Fig. S4) and nitrogen metabolisms (Fig. 2; Table 5) had globally higher relative transcript levels in the 0% NaCl biofilm cultures (Fig. 2).

Higher relative transcript levels (5- to 8-times) for the nap, nor and nos operons were observed in the 0% NaCl biofilm cultures (Table 5). nirK was highly upregulated in the biofilm cultures with 49-fold increase in the relative transcript levels (Table 5). The napGH operon however had a 9.4-fold decrease in the relative transcript levels in the 0% NaCl biofilm cultures. As observed with strain GP59, substantial changes in the relative transcript levels of genes involved in the ammonium transport and the assimilatory NO3−/NO2− reductase were observed with 3- to 22-fold increases in these levels (Table 5, Fig. 4). These results correlate with the absence of NH4+ in the ASW medium, and thus NO3− the only source of N, compared to the 337a medium used for the planktonic pure cultures, which contains 3.8 mM NH4+.

Figure 4 Relative gene expression profiles of selected metabolic pathways of H. nitrativorans strain NL23 in the 0% NaCl biofilm cultures.

The pathways are based on functions deduced by the annotations (provided by KEGG BlastKoala, RAST and GenBank). The arrow thickness is proportional to the value of the ratio of the biofilm-culture TPM divided by the planktonic pure-culture TPM. The blue arrows represent genes with at least 2-fold lower relative transcript levels in the biofilm cultures. The red arrows represent genes with at least 2-fold higher relative transcript levels in the biofilm cultures. The black arrows represent no changes between both types of cultures in the relative transcript levels. The two-component systems and the transporters that are illustrated in blue are encoded by strains GP59 and NL23. See Documents S9 and S10 for gene description.

Figure 4 illustrates changes in relative expression profiles of major pathways in the 0% NaCl biofilm cultures of strain NL23. The relative transcript levels of mxaFJGI increased by 2 folds in the biofilm cultures. The relative transcript levels of the mau operon (methylamine dehydrogenase) showed a 15-fold decrease in the biofilm cultures. The nature of such decrease is unknown as strain NL23 was not fed with methylamine in any of our cultures. The three xoxF genes did not show substantial changes in their transcript levels in both types of cultures. Genes involved in the formaldehyde metabolism to formate and CO2, glycolysis, the tricarboxylic acid cycle, and the pentose pathway showed their transcript levels in general unchanged between the planktonic pure cultures and the biofilm cultures. Few genes in these pathways had 2- to 5-fold differences in their relative transcript levels. The two genes encoding the key enzymes in the serine pathway (alanine-glyoxylate transaminase, glycine hydroxymethyltransferase) had 3- to 7-fold increases in their relative transcript levels in the biofilm cultures. As Methylorubrum extorquens, the NL23 genome encodes the ethymalonyl-CoA pathway (Chistoserdova et al., 2003; Peyraud et al., 2009), which did not show changes overall in the transcript levels of the corresponding genes between the two types of cultures. Contrary to M. extorquens however, a gene encoding the isocitrate lyase is present in strain NL23 and showed a 25-fold upregulation in the biofilm cultures. The isocitrate lyase is one of the key enzymes of the glyoxylate bypass that catalyzes the transformation of isocitrate to succinate and glyoxylate. Gene encoding isocitrate lyase is also present in other available Hyphomicrobium genomes. All these results suggest that in the 0% NaCl biofilm cultures, the carbon metabolism increased in activity and that the glycine regeneration for the serine pathway by the glyoxylate was upregulated. Among genes involved in the oxidative phosphorylation, the relative transcript levels were higher (2–13 times) in the biofilm cultures with those encoding the NADH dehydrogenase, the cytochrome c reductase, with one of the cytochromes c and the F-type ATPase. Combined with increases in the relative transcript levels of the denitrification and the carbon pathways, these results suggest that increases in electron donor activities correlates with the need of electron for the nitrogen dissimilatory metabolism in the biofilm cultures. Strain NL23 possesses four types of cytochrome oxidase (aa3, bo, bd-I and cbb3) (reduction of O2 in H2O) that were about expressed at the same levels in both types of cultures. Figure 4 also illustrates the dynamic changes of transporters and two component systems. Several of these transporters had lower relative transcript levels in the 0% NaCl biofilm cultures. Contrary to strain GP59, genes involved in iron transport were not strongly affected in their gene expression in the 0% NaCl biofilm cultures (Figs. 2 and 4).

The composition of the active microbial community in the biofilm cultures

As mentioned above, the reads from the three metatranscriptomes that did not align with the genomes of H. nitrativorans strain NL23 and M. nitratireducenticrescens strain GP59 and strain JAM1 were de novo assembled. These reads were subsequently aligned to the de novo assembled transcripts to derive the relative levels of these transcripts in the biofilm cultures. The de novo assembled sequences were then annotated for function and affiliation. Finally, these sequences were grouped by microbial affiliation to determine the active populations in the biofilm cultures and to assess their level of involvement in these biofilm cultures (Table 6).

It was estimated that between 5 and 10% reads of the three metatranscriptomes were derived from other microorganisms than H. nitrativorans strain NL23 and M. nitratireducenticrescens strain GP59 and strain JAM1 (Document S4). The proportions of transcripts affiliated to Archaea and Eukarya accounted together for <0.1% (Table 6), which suggests very low abundance of these microorganisms in the biofilm cultures. The proportions of transcripts affiliated to viruses, phages and plasmids in the de novo assembled transcripts represented between 0.6 and 21.7% (Table 6).

Twenty-seven bacterial taxa were selected for their overall transcript levels in at least one of the three biofilm cultures (Table 6). All the taxa detected by the 16S rRNA gene sequencing are present in this list (Table 2 and Document S2). These 27 taxa represented between 22% and 35% of the de novo assembled transcripts. Among these taxa, genes encoding the four denitrification reductases were present in the de novo transcripts affiliated to Marinobacter spp., Stappia spp. and Pseudomonas spp. However, only de novo assembled transcripts affiliated to Stappia spp. showed the complete set of denitrification genes in the three biofilm cultures and organized in operons (napABC, napADFE, norCBQD, nosRZDF).

Further analysis of the expression profiles of the 27 bacterial taxa was performed to assess whether some taxa were influenced in their global metabolic activities by the specific conditions of the biofilm cultures. The overall transcript levels of the 27 bacterial taxa (Table 6) were compared between each biofilm culture by clustering analysis (Fig. 5). NaCl concentration was the main factor of clustering as two distinct clusters were derived. The low salt cluster consisting of nine bacterial taxa showed higher relative transcript levels in the 0% NaCl biofilm cultures, whereas the marine cluster of 13 bacterial taxa had higher relative transcript levels in the Ref300N-23C and 900-30C biofilm cultures. A third cluster showed five bacterial taxa with lower relative transcript levels in the 900N-30C biofilm cultures compared to the 0% NaCl and Ref300N-23C biofilm cultures. In these cases, higher temperature (30 °C vs 23 °C) and higher NO3− and methanol concentration (64.3 mM vs 21.4 mM NO3−; 0.45% vs 0.15% methanol) may have negatively affected these populations.

Table 6 Microbial diversity and the associated denitrification genes in the biofilm cultures from the de novo assembled transcripts.

Affiliation	Biofilm cultures (TPM)	Denitrification genes	
	Ref300N-23C	900N-30C	0%NaCl		
•Actinobacteria					
Streptomyces	6,103	729	2,499	–	
•Bacteroidetes					
Sunxiuqinia	212	504	2,639	norB	
Geofilum	77	109	1,1870	–	
Aequorivita	18,218	25,569	1,398	nirK, norB;C;D;Q, nosZ;DFY	
Winogradskyella	8182	7,917	169	nirK, norC, nosZ;L;D	
Lentimicrobium	3032	3,370	1,671	–	
Xanthomarina	190	4,555	9,598	nirK, norB;D;C;Q, nosZ;L;D	
•Ignavibacteriales	5,513	1,4206	171	napC;H	
•Tenericutes	6,367	5,003	13,409	–	
•Alphaproteobacteria					
Aminobacter	18	438	5,385	napA	
Aquamicrobium	23	456	22,835	napA	
Hoeflea	3,882	87	1,365	narG;H;J;I, nosZ;R;D, nirK	
Hyphomicrobium	0	1,679	3,0057	narH;I, norB;C;Q;D;E, nosZ;D;R;F	
Mesorhizobium	411	1,050	12,505	napA;D;E, nirK, nosZ	
Paracoccus	1,427	1,219	1,204	nirS, narC;D;Q, nosR	
Roseovarius	1,2367	16,479	686	nirS, nirK, norB;C;D;Q;E, nosZ;D;R	
Stappia	39,926	30,885	39,745	napABC, napADFE-nnrS, nirK, norCBQD, nosRZDF;E	
Maritimibacter	6,363	8,360	10,000	narI	
Oceanibaculum	18,115	9,071	25,544	narG;H;J;I	
•Betaproteobacteria					
Azoarcus	48	120	2,854	napA, nosZ	
•Deltaproteobacteria					
Bradymonas	73	9,652	11,178	–	
•Gammaproteobacteria					
Marinicella	36,554	125,466	149,24	nirS, norB;C;Q;D	
Marinobacter	14,545	1,122	264	narG;H;J;I, norB;C;Q, nirK, nirS, nosZ;F;D;R	
Methylophaga	64,802	78,566	80,482	narG;H;J;I, norB;C;D;Q;E, nosD;R	
Idiomarina	5,147	1,027	513	narG;H;J nirK, nirS, norB	
Pseudomonas	6,175	1,333	18,198	narG;H;J;I, nirK, nirS, norB;C;D;Q, nosZ;R,	
Wenzhouxiangella	132	2,857	83	–	
• Other bacteria	113,200	243,523	155,554	a	
• Archaea	113	297	303		
• Eukarya	181	352	416		
• Phages, viruses, plasmids	99,623	5,740	217,095		
• Unclassified	281,015	207,759	157,515		
• Transcripts with no genes	248,782	196,062	148,792		
Notes.

Reads that did not align to the three reference genomes and plasmids were de novo assembled. These reads were then aligned to these assembled sequences. The relative transcript levels of the assembled sequences in a metatranscriptome were expressed as transcripts per million (TPM). Putative genes from the assembled sequences were annotated for function and affiliation. The TPM of the genes affiliated to specific bacterial taxa were then summed. Denitrification genes identified by annotations from respective affiliated bacterial taxa were sorted out.

a Denitrification genes were found scattered in other bacterial taxa.

Figure 5 Hierarchical clustering of selected bacterial taxa in biofilm culture metatranscriptomes.

Heatmap represents differences in the overall gene expression patterns (expressed as TPM; from Table 6) (log10 [TPM by geometric average of TPM]) between the three biofilm cultures for the respective bacterial taxa. Analysis was performed at ClustVis web site (https://biit.cs.ut.ee/clustvis/) (Metsalu & Vilo, 2015).

Discussion

In the environment, numerous bacteria belonging to different taxa can accomplish denitrifying activities, and many of them were encountered in different types of denitrification processes (Lu, Chandran & Stensel, 2014). Very few studies describing the microbial community of methanol-fed denitrification systems have been reported so far. Most of these studies are based on cloned 16S rRNA gene sequences of around 100 clones or based on fluorescence in situ hybridization (Baytshtok et al., 2008; Ginige et al., 2004; Hallin et al., 2006; Neef et al., 1996; Osaka et al., 2008; Osaka et al., 2006; Rissanen et al., 2016; Rissanen et al., 2017; Sun et al., 2016; Yoshie et al., 2006). In all these studies, high proportions of Hyphomicrobium spp. were found in combination with high proportions of other methylotrophs such as Methyloversatilis spp., Methylophilus spp., Methylotenera spp. or Paracoccus spp. The Biodome marine denitrification system showed no exception to this trend with co-occurrence of Hyphomicrobium spp. and the marine methylotroph Methylophaga spp. This co-occurrence was also observed in two other denitrification systems treating saline effluents (Osaka et al., 2006; Rissanen et al., 2016) (see ‘Discussion’ by Payette et al. (2019)). The bacterial diversity of the biofilm taken from the Biodome denitrification system was assessed before when the reactor was operational in 2002 by deriving a 16S rRNA gene library and by culture approach (Labbé et al., 2003; Labbé, Parent & Villemur, 2004). Beside Hyphomicrobium sp. and Methylophaga sp., Paracoccus sp., Sulfitobacter sp., Nitratireductor aquibiodomus, and Delftia sp. among others were identified. In the present report, a more complete determination of the composition of the bacterial community of the denitrifying biofilm that was frozen in 2006 (when the denitrification system was dismantle by the Biodome) was possible with the new sequencing technology.

The composition of the bacterial community of the OB and the IO biofilm cultures was derived from a different region of the 16S rRNA genes (V6–V8) than that used for the other biofilm cultures (V1–V3), and each region was sequenced by a different technology (Illumina and pyrosequencing, respectively). Pyrosequencing technology was no longer available at the time of the sampling of the OB and the IO biofilm cultures, which were carried two years later than the other biofilm cultures. Despites these differences, we believe that the results generated by these two approaches were comparable because they are consistent with the results obtained by qPCR assays (Table 2) that have determined the concentrations of M. nitratireducenticrescens (copies narG1 per ng biofilm DNA) and H. nitrativorans strain NL23 (copies napA per ng biofilm DNA) in the biofilm cultures (Payette et al., 2019). For instance, qPCR showed very low levels of H. nitrativorans strain NL23 in the Ref300C-23C biofilm cultures and high level in the OB. These results concur with the absence of 16S rRNA sequences associated to Hyphomicrobium spp. in the Ref300C-23C biofilm cultures (pyrosequencing) and high number of these sequences in the OB (Illumina sequencing).

The OB cultivated under the different conditions in the ASW medium showed important changes in the Methylophaga and the Hyphomicrobium populations. The proportion of 16S rRNA gene sequences associated to Methylophaga spp. was 3.5% in the OB but was very high, between 85% and 97%, in these cultures. On the contrary, the proportion of 16S rRNA gene sequences associated to Hyphomicrobium spp. was high (46%) in the OB, but was very low (0% to 11%) in these cultures. The NaCl concentrations in the ASW had an impact on the Hyphomicrobium populations. 16S rRNA gene sequences associated to Hyphomicrobium spp. were absent in the biofilm cultures cultivated in ASW at 2.75% NaCl (Ref300N-23C, 300N-30C, 900N-23C, 900N-30C), but were present in the biofilm cultures cultivated at low NaCl concentrations (0%, 0.5% and 1%). These results concur with those obtained by qPCR for the concentrations of M. nitratireducenticrescens and H. nitrativorans strain NL23 (Payette et al., 2019). Cultivating the OB in the IO medium had a different impact on the Hyphomicrobium populations. Although the concentrations of salts in the IO medium and in the ASW with 2.75% NaCl were similar (around 3.5%), persistence of Hyphomicrobium spp. occurred in the IO biofilm cultures. In fact, the concentrations of H. nitrativorans strain NL23 determined by qPCR between the OB and the IO biofilm cultures were at the same levels of magnitude. The lower proportion of 16S rRNA gene sequences associated to Hyphomicrobium spp. in the IO biofilm cultures compared to OB was a consequence of the substantial growth of M. nitratireducenticrescens in these cultures, with a 10-fold increase in concentration as revealed by qPCR (Table 2) (see Payette et al., 2019 for further discussion). Grob et al. (2015) also observed strong growth of Methylophaga spp. in their seawater samples that were fed with 100 µM methanol with relative proportions of 16S rRNA gene sequences raising from <0.5% at T = 0 to 84% after 3 days.

Cultivating the OB in higher concentrations of NO3− and methanol (64.3 mM/0.45%; C/N 1.5) and/or at 30 °C (300N-30C, 900N-23C and 900N-30C biofilm cultures) resulted in increases of 20% to 85% in the specific denitrification rates compared to the Ref300N-23C biofilm cultures (Table 1) (Payette et al., 2019). Temperature was shown to be the main factor of these increases. However, raising the NO3− and methanol concentrations or/and temperature in these cultures did not have an important impact on the bacterial community when compared to the Ref300N-23C biofilm cultures. Metatranscriptomic analysis of the Ref300N-23C and 900-30C biofilm cultures did not reveal either substantial changes in the gene expression profiles between these two cultures. The higher specific denitrification rates of these biofilm cultures could be related to higher metabolisms at the protein levels such as the processing of NO3− by the reductases and transporters.

Results from the 16S rRNA gene sequences showed that Marinicella spp. were present in the OB and all the biofilm cultures, and were the second most abundant bacterial population in the biofilm cultures cultivated in ASW at 2.75% NaCl. These results concur with the metatranscriptomes of the biofilm cultures where Marinicella spp. had the relative transcript levels among the highest in the de novo assembled transcripts. Marinicella spp. are considered strict aerobic bacteria with no indication of NO3− reduction (Romanenko et al., 2010) although a previous study reported high relative abundances of Marinicella spp. in anoxic sulfide oxidizing reactors in which nitrate was used as the electron acceptor (Huang et al., 2015). Genome annotations of two Marinicella strains (GenBank: Marinicella sp. F2—assembly number ASM200005v1 and M. litoralis KMM 3900—ASM259191v1) (Wang et al., 2018) did not reveal complete denitrification pathway, beside a nitric oxide reductase gene cluster also detected here in our metatranscriptomes. Together with the presence of nirS gene (Table 6), these results indicated that Marinicella spp. might have the capacity to use intermediates of the denitrification cycle to support their growth.

The 16S rRNA gene sequences provided evidence of the presence of Pseudomonas spp., Marinobacter spp., Stappia spp., Paracoccus spp. and Aquamicrobium spp. in the OB and in the biofilm cultures. Some species belonging to these genera were reported to carry denitrification. Isolates affiliated to the genera Pseudomonas, Marinobacter and Paracoccus were recovered from the Ref300N-23C biofilm cultures. The Marinobacter and Paracoccus isolates could perform denitrifying activities and grow under anoxic conditions, whereas the Pseudomonas isolate could only consume NO3−.

Although denitrification genes were found in several of the bacterial populations identified by the metatranscriptomic approach, only transcripts encoding the four denitrification reductases affiliated to Stappia spp. were found in the three examined biofilm cultures. The proportions of Stappia spp. in the 16S rRNA gene sequences of these biofilm cultures ranged between 0.09% and 0.42% (Table 2). Stappia spp. are chemoorganotrophic bacteria found in marine environments (Weber & King, 2007) that can oxidize carbon monoxide. They possess the form I coxL gene encoding the large subunit of carbon monoxide (CO) dehydrogenase. Some also contain a gene for the large subunit of ribulose-1,5-bisphosphate carboxylase (RuBisCO) and may be able to couple CO utilization to CO2 fixation (King, 2003). A coxL gene was found in the de novo assembled transcripts affiliated to Stappia spp., but not RuBisCO. Sequences analogue to transporters for simple and multiple sugars such as xylose and fructose, and acetate were found (Documents S5–S7), which suggest that the Stappia bacteria fed on the biofilm material for carbon sources. Combined with the isolation of denitrifying isolates affiliated to Marinobacter spp. and Paracoccus spp., these results suggest that the biofilm has the potential to adapt to heterotrophic non-methylotrophic environments.

The proportion of 16S rRNA gene sequences associated to Bacteroidetes in the OB and in the eight biofilm cultures ranged from 0.2% to 4.9%, and several genera of this phylum were identified in the three metatranscriptomes. Significant proportions of bacteria affiliated to the Bacteroidetes phylum were also found in other methanol-fed denitrification systems. For instance, 29% of cloned 16S rRNA gene sequences were affiliated to Bacteroidetes in an acclimated activated sludge in a methanol-fed anoxic denitrification process treating a synthetic wastewater with 4% NaCl (Osaka et al., 2008). Isolates affiliated to the Bacteroidetes Aequorivita spp. and Winogradskyella spp. were isolated from the Ref300N-23C biofilm cultures. None of these isolates, however, could sustain growth under denitrifying conditions. Although denitrification genes affiliated to Bacteroidetes genera were found in de novo assembled transcripts, genes encoding all four denitrification reductases were not found to any of them. These results suggest that Bacteroidetes are not involved in denitrification, although they may be involved in some steps of the denitrification pathway.

The metatranscriptomic data provided some insights of specific metabolisms in H. nitrativorans strain NL23 and M. nitratireducenticrescens strain GP59 that were regulated in the biofilm environment. In absence of strain NL23 in the Ref300C-23C and the 900N-30C biofilm cultures, the nirK gene of strain GP59 was upregulated by 5–10 times compared to the planktonic pure cultures that were also cultivated under denitrifying conditions. On the contrary, the relative transcript levels of this gene did not change between the 0% NaCl biofilm cultures and the planktonic pure cultures, while the relative transcript levels of the NL23 nirK were 49 times higher in the 0% NaCl biofilm cultures. These results suggest coordination in the expression of nirK between these two strains in the 0% NaCl biofilm cultures.

The gene clusters encoding the three other denitrification reductases (nap, nor, nos) in strain NL23 showed higher relative transcript levels in the 0% NaCl biofilm cultures. napGH was however down regulated in these biofilm cultures. napGH is located in a separate chromosomic region than the napABCDEF operon. NapGH and NapC have redundant function of transferring electrons to NapB across the membrane. It was proposed that NapC transfers electrons from the menaquinol, whereas NapGH do it from ubiquinol (Simon, 2011). The physiological consequence of napGH transcript decrease in the biofilm is unknown. Observations on napEDABC found in the denitrifier Shewanella denitrificans OS217, and napDAGHB in the respiratory NO3− ammonifier Shewanella oneidensis MR-1 suggest that NapGH is more involved in the ammonification system (Simpson, Richardson & Codd, 2010). Despite the denitrifying conditions applied in both types of cultures, the biofilm environment has induced strong up regulation of denitrification genes in H. nitrativorans strain NL23. This may be in response to the rapid processing of NO3− by M. nitratireducenticrescens strain JAM1/GP59 (Mauffrey, Martineau & Villemur, 2015) that could generate rapidly high level of NO2−, which is toxic for strain NL23.

Conclusion

The OB taken from the Biodome denitrification system underwent substantial changes in its bacterial community when subjected to environmental changes. Cultivating the OB in the homemade ASW medium with different formulations (varying NaCl, NO3− and methanol concentrations, and temperature) or in the commercial IO medium showed much higher proportions of Methylophaga spp. in these biofilm cultures compared to the OB. These results concur with the growth of M. nitratireducenticrescens strain GP59 in these cultures. The population of Hyphomicrobium spp. showed a more complex trend. It was at very low levels in the biofilm cultures cultivated in ASW at 2.75% NaCl, but persisted in the biofilm cultures cultivated in ASW at low NaCl concentration, and also cultivated in the IO medium. Other denitrifiers affiliated to Marinobacter spp. and Paracoccus spp. were isolated from the biofilm cultures. Moreover, metatranscriptomic analysis revealed that denitrifying bacteria affiliated to Stappia spp. were metabolically active in the biofilm cultures. The biofilm environment has favored the upregulation of the denitrification pathway in M. nitratireducenticrescens strain GP59 and H. nitrativorans strain NL23 compared to planktonic pure cultures, despite the facts that these two types of cultures were grown under denitrifying conditions. All these results demonstrated the dynamics and the plasticity of the denitrifying biofilm to sustain environmental changes and illustrate a comprehensive picture of the microbial community of the biofilm and its adaptation to these changes. This could benefit in the development of optimal denitrifying bioprocesses under marine conditions. For instance, the fact that non-methylotrophic, denitrifying bacteria are present in the biofilm could suggest adaptation of the denitrification process to another source of carbon such as ethanol or acetate.

Supplemental Information

Table S1 Composition of natural seawater, the INSTANT OCEAN brand salt and the artificial seaeater (ASW) used in this study

Click here for additional data file.

Tables S2–S3 Hyphomicrobium-and Methylophaga-affiliated OTUs and in the OB and the IO biofilm cultures

Click here for additional data file.

Figure S1 Schematic of the experimental assays performed in the study

Click here for additional data file.

Figure S2 Evolutionary relationships of OTUs derived from the OB and the IO biofilm cultures with the genera Hyphomicrobium and Methylophaga

Click here for additional data file.

Figures S3–S4 Relative expression profiles of M. nitratireducenticrescens strain GP59 and H. nitrativorans Nl23 in the biofilm cultures

Click here for additional data file.

Supplemental Information 1 Supplemental data summary

Brief description of: the 14 supplemental documents, the accession numbers of the 16S rRNA sequences and transcriptomes, and where they were used to derive the Figures, Tables and supplemental files.

Click here for additional data file.

Document S1 RTL-Data_Analysis_Methodology

Procedures used by the RTL sequencing service to process the 16S rRNA sequence reads.

Click here for additional data file.

Document S2 Raw data for Fig. 1 and Table 2, and OTU sequences

Raw data of Figure 1: number and proportion of 16S rRNA sequence reads derived from the biofilm cultures. Proportion of 16 rRNA sequence reads associated to bacterial taxa, and used for Principal Component Analysis.

Raw data of Table 2: number of 16S rRNA sequence reads per identified taxon derived from the biofilm cultures.

List of OTUs with their affiliation, cluster size and sequence in the biofilm cultures.

Click here for additional data file.

Document S3 Metatranscriptome analysis

Reference genome: strain GP59 Biofilm cultures-vs-pure cultures

Reference genome: strain NL23 Biofilm cultures-vs-pure cultures

Reference genome: strain JAM1 Biofilm cultures-vs-pure cultures

Click here for additional data file.

Document S4 Raw data for Table 6

Affiliation of the associated genes found in the de novo assembled sequences

Estimation of the proportions of the microbial taxa in the metatranscriptomes

Denitrification genes found in selected bacterial taxa from the de novo assembly

Click here for additional data file.

Document S5 Analysis of de novo assembled transcripts of the Ref300N-23C biofilm cultures

Click here for additional data file.

Document S6 Analysis of de novo assembled transcripts of the 900N-30C biofilm cultures

Click here for additional data file.

Document S7 Analysis of de novo assembled transcripts of the 0% NaCl biofilm cultures

Click here for additional data file.

Document S8 Raw data for Fig. 2 and Figs. S3–S4. Supplemental information regarding the metatranscriptome analysis associated to strain JAM1

Click here for additional data file.

Document S9 Raw data for Fig. 3A, Table 5 and Figs. S3–S4 related to strain GP59

Click here for additional data file.

Document S10 Raw data for Fig. 3B, Table 5 and Figs. S4 related to strain NL23

Click here for additional data file.

Document S11 Metatranscriptome analysis associated to strain JAM1

Click here for additional data file.

Document S12 Raw data for Fig. 4

log10 [Taxa TPM/geometric average of TPM]

Click here for additional data file.

Document S13 Aligned sequences for Fig. S2A (Fasta format)

Click here for additional data file.

Document S14 Aligned sequences for Fig. S2B (Fasta format)

Click here for additional data file.

We thank Karla Vasquez for her technical assistance.

Additional Information and Declarations

Competing Interests

Author Contributions

DNA Deposition

Data Availability

The authors declare there are no competing interests.

Richard Villemur conceived and designed the experiments, performed the experiments, analyzed the data, contributed reagents/materials/analysis tools, prepared figures and/or tables, authored or reviewed drafts of the paper, approved the final draft.

Geneviève Payette and Valérie Geoffroy conceived and designed the experiments, performed the experiments, analyzed the data, prepared figures and/or tables, authored or reviewed drafts of the paper, approved the final draft.

Florian Mauffrey analyzed the data, authored or reviewed drafts of the paper, approved the final draft.

Christine Martineau conceived and designed the experiments, authored or reviewed drafts of the paper, approved the final draft.

The following information was supplied regarding the deposition of DNA sequences:

16S rRNA sequence reads are available in the GenBank Sequence Read Archive (SRA) under the accession number PRJNA524642.

The 16S rRNA gene sequences are available in GenBank under the accession numbers MK571459–MK571476.

RNAseq reads from the planktonic pure cultures and the biofilm cultures are available in the SRA under the accession number PRJNA525230.

De novo assembled transcripts are available in SRA under the accession number PRJNA525230.

Annotations of the de novo assembled transcripts are available at the Joint Genomic Institute (https://img.jgi.doe.gov/cgi-bin/m/main.cgi) under the GOLD Analysis Project Id: Ga0307915, Ga0307877, Ga0307760.

The following information was supplied regarding data availability:

Raw data is available in the Supplemental Files.

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
