# Peer review of "Dynamics of a methanol-fed marine denitrifying biofilm: 2—impact of environmental changes on the microbial community"

_PeerJ, doi:10.7717/peerj.7467_

## Round 0.1 · original submission · Major Revisions

Dear Dr. Villemur and colleagues:

Thanks for submitting your manuscript to PeerJ. I have now received three independent reviews of your work, and as you will see, the reviewers raised some major concerns about the research. Despite this, these reviewers are optimistic about your work and the potential impact it will have on the metagenomics of methanol-fed, denitrifying marine biofilm. Thus, I encourage you to revise your manuscript, accordingly, taking into account all of the concerns raised by all reviewers.

While the concerns of the reviewers are relatively minor, this is a major revision to ensure that the original reviewers have a chance to evaluate your responses to their concerns.

I look forward to seeing your revision, and thanks again for submitting your work to PeerJ.

Good luck with your revision,

-joe

Reviewer 1 ·

Basic reporting

The authors conducted meta-transcriptome analysis to compare expressing gene and potentially growing bacteria strain among different conditions of two temperatures, four saline conditions, and four nitrate concentrations. The control experiments were performed and were well summarized as showing high quality figures and tables. Additionally, they also isolated some bacteria strain from the biofilm from denitrification reactor in a national science museum. However, the manuscript includes many ambiguous contexts and was written unclearly. The inconsistence between objectives and conclusion is critical, besides the objectives is unclear in the manuscript. The text format and wording, especially technical terms, should be collected.

Experimental design

Each experiment was designed well and well written. However, the objectives and conclusion were not clearly written.

Validity of the findings

The objectives of this study were stated in abstract (L27–28), ... impact of culture conditions on the dynamics of the overall microbial community.... However, it is not mentioned in the introduction section where "objectives" should be explained logically and clearly. Worse, the answer for the objectives were unclearly mentioned in the conclusion section. It is unclear that what the impact of NaCl, nitrate and temperature on the denitrifying biofilm.

The sections of introduction and results includes inappropriate descriptions.
For introduction section, information of the biofilm from the Biodome is well written but the appropriate explanations for meta-transcriptome analysis (L112–115) and isolation of bacteria strains (L116–177) were lack. Worse, the last paragraph showed the results of the present study that should be in results section.
For results section, some paragraphs include the descriptions of materials and methods, eg. L231–237, L286–296, L304–314, L369–373, and L428–434. Please re-construct the context throughout the manuscript to make descriptions in appropriate section.

Additional comments

Ambiguous and unknown wording should be corrected.
For the bacteria strain of M. Nitratireducenticrescens strain GP59, some different descriptions were seen, eg. M. Nitratireducenticrescens strain GP59, strain GP59, and GP59. Please use appropriate term for them.
In addition, the description of "GP59 gene" is not correct word for indicating "genes of strain GP59".

Unknown/Undefined terms: "concentration". (L92 and L303)

Please correct the format of references in text. (L469–471)

Reviewer 2 ·

Basic reporting

The manuscript was clearly written and well structured, however some language errors (spelling and grammar) were found throughout the text. Please proofread the text.

I understand this study is one in a series of studies, however, this paper should be self standing without knowledge of the previous work or even more, work that has has been done but not published yet. Throughout the paper there are references to a submitted article (Payette et al 2019) which is not accessible to the reviewer at this point. Please provide sufficient information in the text.

Figures and tables are relevant to the article.

Raw sequence data were properly deposited in public repositories.

Experimental design

The article is within the aims and scope of the journal.

In this study, the authors studied the composition of microbial community of a denitrifying biofilm maintained under different growth conditions, studied the transcriptional activity of bacteria in a denitrifying biofilm and in planktonic cultures. The authors also isolated denitrifying bacteria from the biofilms into pure cultures. As stated in lines 599-600, the information from this study should be applicable for development of an an optimal denitrification bioprocess for a marine system.

There was a lot going on in this paper and significant efforts were put in to produce the data. However, I think the study should be framed better, the main goals for this particular paper should be be more clearly stated and the results should be presented so that they form a coherent story and answer the specific research questions. In its present form, different parts seem to be loosely tied together and the results seem more descriptive than answering the specific research questions or proofing/disproofing hypotheses.

One of my major criticisms regards the experimental design. In this study, the authors wanted to test the effect of different environmental conditions on microbial community composition and gene expression patterns. The conditions for the cultures were listed in Table 1. According to this information, in culture 900N-23C two parameters and in 900N-30C three parameters are changed simultaneously compared to the control. How is it possible to to tell apart the effect of a single parameter when multiple parameters are changed at once?

There were also some major shortcomings in describing the methods: The reporting of bioinformatics methods (Lines 146-150) for the 16S reads is not sufficient to replicate the analysis. You should report which programs and which versions were used at each step. Were there any changes to default parameters? Also references to computer programs, such as UCHIME, are missing. Secondly, from line 123 onwards in materials and methods-section: there are some qPCR data in results but the qPCR assay is not described in materials and methods.

Validity of the findings

The conclusions starting from line 580 seem rather general and they didn't really answer all of the specific research questions presented in the beginning of the article.

There was no statistical testing for the microbial community composition data to tell whether the observed changes were significant or not?

Overall, in the light of the results and conclusions, it remained unclear, how do these results help in developing better denitrification systems for salt water systems.

Additional comments

Below are some detailed comments regarding some specific issues in the text.

Lines 23-24: Line "...exposed to a range of physico-chemical parameters" sounds odd as you probably mean environmental conditions. Please reword the sentence.
Line 74: Please add a reference for this estimate.
126-128: the reference here is not accessible and it is not possible for a reviewer to understand the difference in composition of ASW and IO media. They should be described here briefly.
Lines 129, 166 and elsewhere: is "scrapped" correct or should it be "scraped"?
Lines 146-154: I'm afraid thet the use of 16S primers targeting different regions and two different sequencing technologies does not provide data that would allow a valid comparison between samples.
Line 150: how were the samples stored before sending to sequencing? Did they stay frozen the entire time? I'm afraid that the long storage could affect the sequencing outcome as well. Especially if the samples were handled during the storage.
Lines 208-210: how was the statistical analysis performed if there were no replicates?
Lines 244, 246: I think it is confusing to use therm "increased" and "decreased" here as the results came from two separate cultures.
Lines 248-249: According to Table 1 also methanol concentration varied.
Lines 450-460: Considering the goals of the study, why are the overall transcript abundances relevant?
Lines 492-494: I don't understand the relevance of the qPCR data in table 2 and how the qPCR targeting the narG1 gene would solve the issue (as stated on lines 492-494), especially when the reference (marked in the list as submitted) is not accessible at this point. Please clarify.
Lines 595-596: unclear sentence "and (an) uncultured denitrifier(s)...", please revise.

Figure 2. For clarity, you should explain what TPM means (in the figure legend)
Figure 4: The figure legend should express more clearly that the clustering is based on overall gene expression patterns.

·

Basic reporting

Villemur et al. provide a manuscript in which they assess the influence of NaCl, nitrate and temperature on the denitrifying community in a methanol-fed marine biofilm. The main drivers in this marine denitrifying biofilm are Methylophaga nitratireducenticrescens and Hyphomicrobium nitrativorans. This study complements two other studies. In this study, they are able to show, that this biofilm can adapt to heterotrophic, non-methylotrophic conditions.

Villemur et al. provide a clear introduction and background to the topic. However, two important concerns came up reading this manuscript:
1) The authors carried out 16S rRNA gene sequencing, however called it throughout the manuscript “metagenomics” or “metagenomes”. Referring to 16S rRNA gene surveys as "metagenomics" is misleading. Here and elsewhere, do please not confuse or conflate amplicon sequencing and metagenomics. They are not the same.
2) The authors state e.g. in line 239 and 488 that they used 16S rRNA gene sequencing in order to assess the evolution of the bacterial community. However, throughout the manuscript I could not find any analysis assessing the evolution of the bacterial community, such as e.g. ancestral state reconstruction analysis or others. Maybe, it would be clearer to say that they used 16S rRNA gene sequencing in order to assess the phylogeny of the bacterial community.


Villemur et al. provide clear and relevant figures of high-quality.

In line 41 it is not clear what is meant with “it”. Is it Methylophaga or Hyphomicrobium or both? Please clarify.


Following language changes are suggested:

Lines 26/27: Here, we report the impact of these culture conditions on…

Lines 34/141/230/443/484/489/497/507/521/531/543: Please don’t use metagenomics or metagenomes for 16S rRNA gene sequencing.

Line 36: metatranscriptomes of selected biofilm cultures

Line 40: in the OB, both were absent

Line 43: In these biofilm cultures, emergence of Marinicella sp. occurred.

Line 54: It is suggested to delete “in the biofilm” from this sentence, to allow a better flow of the sentence.

Lines 65/66: of the bioprocesses than a single species biofilm

Line 74: to be composed of

Line 101: under batch mode and anoxic conditions

Line 103: and varying temperatures

Line 108: It is suggested to use “investigated” instead of measured in this sentence.

Lines 115/116: Finally, a culture dependent approach

Line 120: Our results demonstrate

Line 135: change here and elsewhere: “the Reference biofilm” to “the reference biofilm”

Lines 171/180: start Methylophaga with a capital letter and please in italics (here and elsewhere)

Line 230: Bacterial composition of the biofilm cultures by 16S rRNA gene sequencing

Line 239: It is suggested to use phylogeny instead of evolution

Line 241: Here and elsewhere, please do not use “16S". The correct term is “16S rRNA genes” for DNA or "16S rRNA” for RNA.

Line 244: “ca.” does not need to be in italics

Line 293: To assess the contribution

Lines 357/360: Here and elsewhere, please leave a space between % and NaCl

Lines 422/423: It should be either “ had a lower relative transcript level” or “had lower relative transcript levels”

Line 553: encoding all four denitrification reductases were not found in any of them.

Line 690: Paracoccus should be in italics

Line 719: Mainicella litoralis should be in italics

Line 728: Shewanella should be in italics

Line746: Marinicella sediminis should be in italics

Experimental design

The experimental design of this study is sound and clear.

Could the Electropherogram of the total RNA and the rRNA-depleted mRNA be provided into the Supplementary material?

Validity of the findings

Lines 279/280: “The three isolates affiliated to the Paracoccus spp. have identical 16S rRNA gene sequences” – Are they in fact the same species?

In lines 347/348 the authors state that ‘the strong upregulation of this operon may be related to the increasing need of NADH in the biofilm.” Readers might wonder why there is an increasing need for NADH in the biofilm. I would recommend to include one ot two sentences explaining this.

---

## Round 0.2 · Minor Revisions

Dear Dr. Villemur and colleagues:

Thanks for revising your manuscript. The reviewers are very satisfied with your revision (as am I). Great! However, per reviewer 1, there are a few minor edits to make. Please address these ASAP so we may move towards acceptance of your work.

Best,

-joe

Reviewer 2 ·

Basic reporting

In Figure 1, it seems that the panels A and B have flipped in the pdf-version of the manuscript thet was sent to the reviewers. Please make sure that the panels are in correct order. Also, in the Fig. 1B PCA-plot, it would be good to separate the different conditions (IO, OB and ASW-cultures) with different symbols or at leas label the samples originating from cultures as well. Having a group of unlabeled symbols on the right side of the plot is a bit confusing.

Experimental design

No comment

Validity of the findings

No comment

Additional comments

Overall, the authors have improved the manuscript significantly by making revisions suggested by the reviewers. The authors have addressed my comments sufficiently.

·

Basic reporting

All changes were made as suggested and the manuscript was considerably improved.

Experimental design

All changes were made as suggested and the manuscript was considerably improved.

Validity of the findings

All changes were made as suggested and the manuscript was considerably improved.

---

## Round 0.3 · accepted · Accept

Dear Dr. Villemur and colleagues:

Thanks for revising your manuscript to PeerJ, and for addressing the concerns raised by the reviewers. I now believe that your manuscript is suitable for publication. Congratulations! I look forward to seeing this work in print, and I anticipate it being an important resource for research pertaining to the metagenomics of methanol-fed, denitrifying marine biofilm.

Thanks again for choosing PeerJ to publish such important work.

-joe